# A new energy transfer channel from carotenoids to chlorophylls in purple bacteria

Jin Feng[1], Chi-Wei Tseng[2], Tingwei Chen[1], Xia Leng[1], Huabing Yin[3], Yuan-Chung Cheng[2], Michael Rohlfing[4] & Yuchen Ma[1]

It is unclear whether there is an intermediate dark state between the $S_2$ and $S_1$ states of carotenoids. Previous two-dimensional electronic spectroscopy measurements support its existence and its involvement in the energy transfer from carotenoids to chlorophylls, but there is still considerable debate on the origin of this dark state and how it regulates the energy transfer process. Here we use ab initio calculations on excited-state dynamics and simulated two-dimensional electronic spectrum of carotenoids from purple bacteria to provide evidence supporting that the dark state may be assigned to a new $A_g^+$ state. Our calculations also indicate that groups on the conjugation backbone of carotenoids may substantially affect the excited-state levels and the energy transfer process. These results contribute to a better understanding of carotenoid excited states.

[1] School of Chemistry and Chemical Engineering, Shandong University, Jinan 250100, China. [2] Department of Chemistry and Center for Quantum Science and Engineering, National Taiwan University, Taipei 106, Taiwan. [3] School of Physics and Electronics, Henan University, Kaifeng 475004, China. [4] Institut für Festkörpertheorie, Universität Münster, Münster 48149, Germany. Correspondence and requests for materials should be addressed to Y.M. (email: myc@sdu.edu.cn)

Carotenoids (Cars) take part in various processes in living organisms. In living animals and humans, they act as antioxidants preventing free radicals from destroying tissue cells[1] and are relevant to the vision of retina[2]. In the photosynthesis of plants and microorganisms, Cars are responsible for harvesting light, transferring energy to chlorophylls (Chls), and protecting against excessive light by quenching excited states of Chls[3, 4]. Exploring the excited-state dynamic properties of Cars is fundamental to understand the mechanism of photosynthesis and is also helpful for developing artificial light-harvesting systems. Despite the studies for several decades, our knowledge on the electronic structure and excited-state properties of Cars, which determine the mechanism of energy decay in Cars and energy flow from Cars to Chls, is still limited. For example, the origin and nature of a dark state, which lies between the strongly one-photon allowed $S_2$ ($1B_u^+$) state and the forbidden $S_1$ ($2A_g^-$) state, has triggered intense research and has been under debate. This dark state would have a critical role in mediating the Car-to-Chl energy transfer process and the depopulation of the $S_2$ state[4–11]. Discovery of this dark state would make the conventional carotenoid photophysics model $S_0{\rightarrow}S_2{\rightarrow}S_1{\rightarrow}S_0$ (Fig. 1a), where $S_0$ is the ground state $1A_g^-$, no longer accurate. Therefore, a new model may be required to account for the excited-state behavior of Cars[4, 12, 13].

A number of experimental and theoretical studies[4, 8, 14, 15] have been carried out in the past 20 years to unravel this dark state. The $1B_u^-$ state is the popular choice for the attribution of the dark state (Fig. 1b), and it also seems to be the only choice based on the present theory on the excited-state structure of polyenes, which are closely related to the Cars. However, this assignment is controversial due to the contradiction between the properties of the dark state measured experimentally and the behavior of the $1B_u^-$ state determined from both experiments and theory[4, 9]. Especially, the conjugation-length dependence of the $1B_u^-$ state is not in accord with that of the dark state as observed in the emission spectra and transient absorption spectra[16–20]. Recent two-dimensional electronic spectroscopy (2DES) and high-time resolution broadband pump-probe spectroscopy measurements on Cars spheroidene ($N = 10$), rhodopin glucoside (RG) ($N = 11$) and spirilloxanthin ($N = 13$) demonstrate clearly the decay of the $S_2$ state to the dark state and the energy transfer channel from the dark state to the $Q_x$ state of Chls[6, 10, 21, 22]. From these experiments, emission energy of the dark state seems to lie a little bit (~0.1 eV) below that of the $S_2$ state, irrespective of the conjugation length of Cars. Decrease of the $1B_u^-$ energy with conjugation length is much steeper than this dark state. In addition, emission energy of the $1B_u^-$ state might be smaller than 2 eV for Cars with $N = 10$–13 as predicted by Koyama et al.[4, 18], according to which energy transfer to the Chls $Q_x$ state, whose absorption energy is ~2.1 eV, cannot realize. Thus, there may exist some other, unknown excited state in the vicinity of the $S_2$ state.

Another important issue is how much the dark state is involved in the Car-to-Chl energy transfer. There is disagreement in the overall Car-to-Chl energy transfer efficiency between theoretical prediction and experimental measurement, e.g., 20 vs. 50–60% for *Rhodopseudomonas (Rps.) acidophila*[23, 24]. As the spectral overlap of the dark state emission and $Q_x$ absorption is larger than that between the $S_2$ and $Q_x$ states, the dark state, which is not considered in previous theoretical calculations, has been supposed to account for this disagreement[6]. However, some experiments demonstrate that the amount of energy transferred from dark states to Chls is minor[4, 25, 26].

Here, we examine the excited-state dynamics of two Cars from purple bacteria and their Car-to-Chl energy transfer by virtue of many-body Green's function theory and Förster–Dexter theory. 2D spectrum is also simulated by a density matrix-based

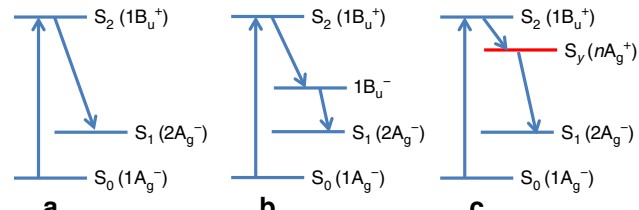

**Fig. 1** Photophysics model of carotenoids. The three-state model (**a**), the four-state model involving $1B_u^-$ (**b**) and the new four-state model containing $S_y$ (**c**) used to interpret the excited-state dynamics of carotenoids

dynamical method to compare with experiments. We provide evidence supporting a new dark state with $A_g^+$ symmetry, which is denoted by $S_y$, in Cars. Its absorption energy would be higher than the $S_2$ state, whereas its emission energy would be lower. The $S_y$ state is a singly excited state, in stark contrast to the well-known dark states $2A_g^-$ and $1B_u^-$ which are doubly excited states constituted by the two triplet excitons. The excited-state structure and the energy transfer appear to be highly sensitive to the methyl groups on the conjugated backbone of Cars.

## Results

**Excitation energy of $S_y$.** Extensive computational researches for the dark state in Cars have been performed using various first-principle methods[8, 14, 27–33]. However, they mainly focus on the controversial $1B_u^-$ state. With many-body Green's function theory, we study the excited-state dynamics of RG ($N = 11$) in *Rps. acidophila* (Fig. 2) and spirilloxanthin ($N = 13$) in *Rhodospirillum rubrum*. The calculated absorption energy of the $S_2$ state for RG and spirilloxanthin are 2.65 and 2.42 eV. In experiments, the $S_2$ energy is 2.48 eV for RG in methanol and 2.36 eV for spirilloxanthin in *n*-hexane[4]. In solution and the protein environment, absorption spectra of Cars redshift owing to the polarizability of the medium[4, 22, 34, 35]. If taking this into account, our calculations agree well with experiments. We also prove that putting some protein fragments near Cars has limited effects on the excitation energies of Cars (Supplementary Fig. 2). Excitation energies of the $S_1$ and $1B_u^-$ states for RG (spirilloxanthin) are calculated to be 1.87 and 2.61 eV (1.68 and 2.32 eV), respectively, employing the scheme proposed by Tavan and Schulten[36]. Above $S_2$ by 0.55 eV for RG and 0.52 eV for spirilloxanthin, our results predict a new state ($S_y$) of the $A_g^+$ symmetry. It is optically forbidden from the ground state and must be indiscernible in experimental optical absorption spectra. The $S_y$ state is a singly excited state, with the wave function represented dominantly by the transitions HOMO $-1 \rightarrow$ LUMO and HOMO $\rightarrow$ LUMO+1 (Figs. 3, 4; Supplementary Fig. 3 and Supplementary Note 2). The high-level quantum chemistry approach EOM-CCSD can also get this state and a similar $S_2$–$S_y$ energy gap (Supplementary Table 2). The transition dipole moment of the $S_y$ state is 0.7 and 1.6 Debye for RG and spirilloxanthin, respectively, much smaller than that of the $S_2$ state (20.3 and 28.6 Debye) and comparable to the dark state measured in experiments[18]. Moreover, the $S_y$ state remains optically forbidden when twisting the conjugated backbone of Cars from all-*trans* to *cis* configurations (Supplementary Fig. 4 and Supplementary Table 3).

**Emission energy of $S_y$.** After optical absorption, Cars first relax in the $S_2$ state. From the ground-state geometry ($R_0$ in Fig. 3) to the excited-state minimum geometry of the $S_2$ state ($R_2$ in Fig. 3), Stokes shift of the $S_2$ state is 0.18 eV for both RG and spirilloxanthin. This is in accord with the shift of 0.15 eV for RG measured experimentally[4, 23]. In this process, the $S_y$ state downshifts

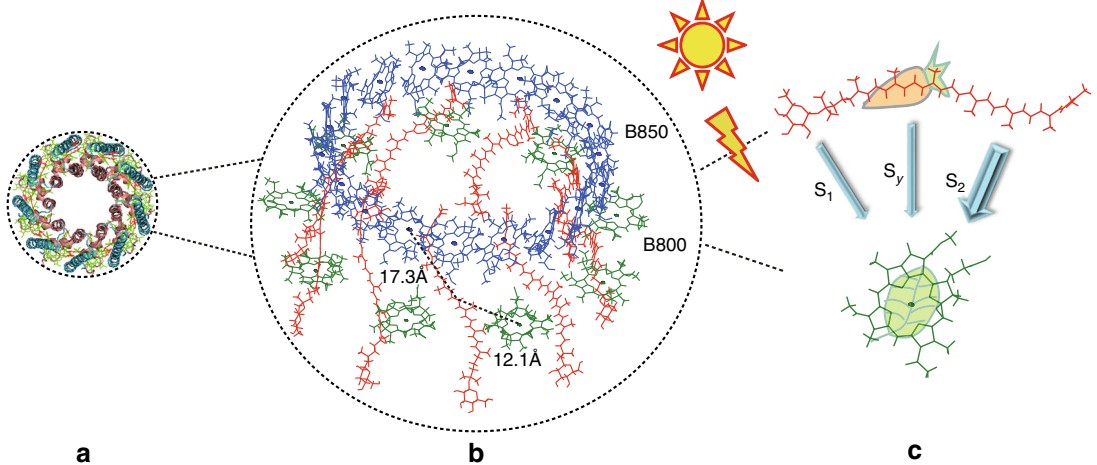

**Fig. 2** Structure of *Rps. acidophila*. **a** Top view of the structure of light-harvesting complex II in *Rps. acidophila*. **b** Enlarged view of carotenoids and chlorophylls in **a**. For a better distinction, carotenoids, the nine B800 chlorophylls, and the eighteen B850 chlorophylls are depicted in *red*, *green*, and *blue*, respectively in **b**. After these antenna pigments absorb light, carotenoids transfer energy to chlorophylls B800 and B850, mainly through the $S_2$ state, partly through the $S_y$ and $S_1$ states as illustrated in **c**

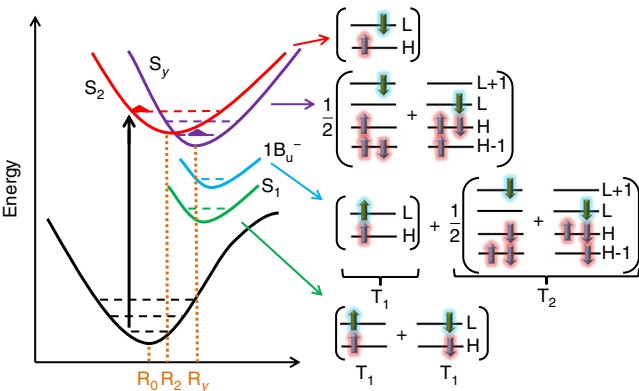

**Fig. 3** Scheme of the potential energy curves for carotenoids with $N \geq 9$ (*left*) and the compositions of excited states (*right*). $R_0$, $R_2$, and $R_y$ represent geometries at the potential minima of states $S_0$, $S_2$, and $S_y$, respectively. H and L represent the highest occupied and the lowest unoccupied molecular orbitals, respectively. *Arrows* on the orbital illustrate the population and the spins of electrons in this orbital. $S_2$ and $S_y$ are singly excited states. $S_1$ (formed by two $T_1$) and $1B_u^-$ (formed by $T_1$ and $T_2$) are doubly excited states. $T_1$ and $T_2$ are the lowest and the second lowest triplet exciton, respectively

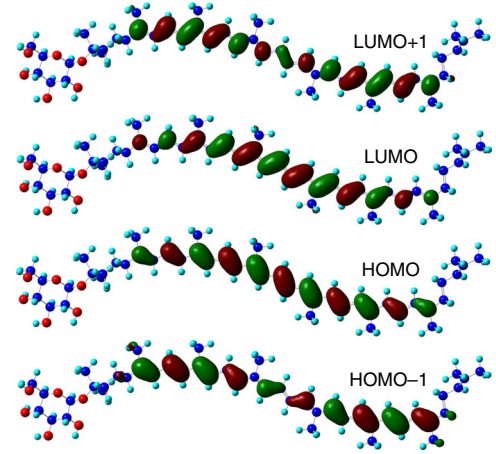

**Fig. 4** Molecular orbitals of rhodopsin glucoside. The wave function isosurfaces for the highest two occupied and the lowest two unoccupied orbitals of rhodopsin glucoside. C, O, and H atoms are shown in *blue*, *red*, and *cyan*, respectively

by 0.66 eV for RG and 0.68 eV for spirilloxanthin, respectively (Fig. 3). Now, at the potential minimum of the $S_2$ state, the energy of the $S_y$ state is just a little bit higher (0.07 eV for RG and 0.02 eV for spirilloxanthin) than the $S_2$ state. If extrapolating further along the $R_0 \rightarrow R_2$ reaction coordinate, the energy of the $S_y$ state falls by an additional 0.1 eV (Fig. 3). The emission energy of the $S_y$ state (at $R_y$ in Fig. 3) is thus lower than that of the $S_2$ state (see also Supplementary Figs. 8 and 9; Supplementary Note 4). Although the crossing point between the $S_2$ and $S_y$ states is not in the $R_0 \rightarrow R_2$ region, nonadiabatic transition from $S_2$ to $S_y$ can happen with the aid of vibration. The $S_y$ state may also have a role in tuning the relaxation to lower states like $S_1$ (Fig. 1c). The emission energy of the $S_y$ state for RG and spirilloxanthin is above the absorption energy of the chlorophyll $Q_x$ state, making the Car-to-Chl energy transfer via the $S_y$ state realizable.

From the $R_0$ to $R_2$ geometries, the bond length alternation, i.e., the difference between the average lengths of C–C and C=C bonds, is reduced. As the $S_2$ state is an ionic excited state, whereas

the $S_y$ state can be considered to be a covalent-like one due to its weak transition dipole moment, the electron-hole binding energy ($E_b$) in the $S_2$ state should be much more influenced by structural variation than that in the $S_y$ state. We do find that from $R_0$ to $R_2$, $E_b$ in the $S_y$ state remains constant while that in the $S_2$ state reduces by ~0.5 eV. From $R_0$ to $R_2$, the gap between the unoccupied and occupied molecular orbitals ($E_{gap}$) narrows by ~0.7 eV. The excitation energy, which equals $E_{gap} - E_b$ in physics, thus decreases faster for the $S_y$ state than the $S_2$ state from $R_0$ to $R_2$ (Fig. 5c). Energies of the $S_1$ and $1B_u^-$ states are also found to exhibit much higher sensitivity on the bond length alternation than the $S_2$ state[8, 36].

**Dependence of the $S_y$ emission energy on conjugation length.** Dependence of the emission energy of the dark state on the conjugation length of Cars is a crucial factor to determine the attribution of the dark state[4]. Polyenes are ideal models to investigate this issue. We examine a series of polyenes $H_3C-(C_2H_2)_N-CH_3$ with $N = 6-15$ (Fig. 5). Although the absorption energy of the $S_y$ state varies faster with the conjugation length than that of the $S_2$ state (Fig. 5a), the $S_2-S_y$ energy gap at

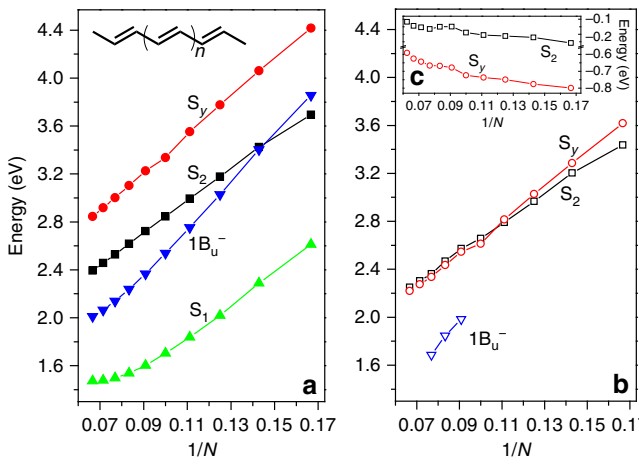

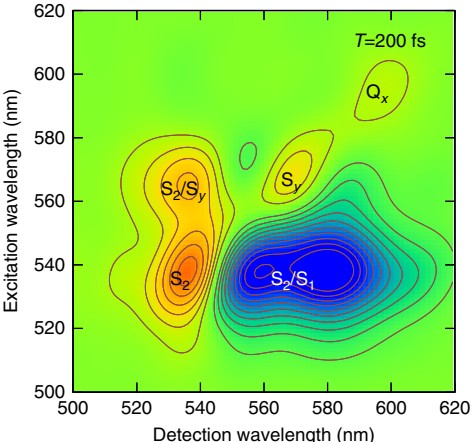

**Fig. 5** Dependence of excitation energies of polyenes on the conjugation length $N$. **a** At the ground-state geometry. **b** At the $S_2$ state potential minimum geometry. **c** The energy shifts of the $S_2$ and $S_y$ states from the ground-state geometry to the $S_2$ state potential minimum geometry. Energies of $1B_u^-$ in **b** are taken from the experimental fluorescence spectra[4]

**Fig. 6** Simulated two-dimensional (2D) electronic spectrum of *Rps. acidophila* at the population time $T = 200$ fs. The diagonal peaks are symbolized by the corresponding state; the cross-peaks indicate that energy is transferred from the state represented by the *first symbol* to the acceptor state represented by the *second symbol*

the potential minimum of the $S_2$ state remains at ~0.03 eV for $N \geq 10$ with the $S_y$ below $S_2$ (Fig. 5b). Taking into account the additional 0.1 eV downshift of the $S_y$ state to its own potential minimum as discussed above, the emission energy of the $S_y$ state is lower than that of the $S_2$ state for $N \geq 9$ and the gap between them remains at ~0.1 eV. This agrees well with the transient absorption studies that the dark state lies below the $S_2$ state for Cars with $N \geq 9$[11], and also 2DES measurements that the potential minimum of the dark state is 0.1 eV below that of the $S_2$ state for $N = 10$, 11, and 13[6, 10, 21].

**2D spectrum of *Rps. acidophila*.** In a recent study, Scholes and coworkers utilized the broadband 2DES to investigate carotenoid dark states in purple bacteria[6]. The 2DES is a four-wave mixing technique that is extremely sensitive to electronic coherence and energy relaxation dynamics[37, 38]. To verify our model, we calculate theoretical 2D spectrum of *Rps. acidophila* based on a model Hamiltonian that is consistent with our ab initio calculations. Note that the original 2D experiment is carried out using a broadband pulse that overlaps with the very red edge of the $S_2$ band and the blue edge of the $Q_x$ band in a sample of *Rps. acidophila*. The experimentally observed 2D $S_2$ peak at ~535 nm (2.34 eV) is dependent of the excitation laser spectrum, and is also in good agreement with the calculated $S_2$ transition energy at the $S_2$ minimum. This excitation energy corresponds to a highly displaced geometry along the bond length-alternation coordinate. In this geometry, the four-state model with the $S_y$ state energy lower than that of the $S_2$ state applies, therefore, we place the $S_y$ energy minimum at 0.1 eV below the $S_2$ energy minimum in our 2D simulation, in accordance with our calculations for RG. Figure 6 shows the simulated 2D spectrum at a delay time of 200 fs, and the simulated spectrum is in agreement with the experimental spectrum[6]. Specifically, the diagonal peaks, $S_2/S_y$ cross-peak, and pronounced $S_2/S_1$ excited-state absorption peak that splits into two by the lower-diagonal $S_2/S_y$ cross-peak are all correctly reproduced in our model simulations. Therefore, our theoretical calculations are consistent with the 2D experiments.

**Influence of groups on the excitation energies of Cars.** Comparing RG with the $N = 11$ polyene and spirilloxanthin with the $N = 13$ polyene, the $S_y$–$S_2$ energy gap in the Cars is the same as that in the polyene. However, the $S_2$–$1B_u^-$ and $S_2$–$S_1$ energy gaps in the polyene are about 0.3 eV wider than those in the Cars

of the same conjugation length. The structural difference between Cars and polyenes is reflected in two respects: (i) the conjugated backbone is symmetrical in polyenes but distorted in Cars, (ii) there are groups, e.g., methyl groups, on the conjugated backbone in Cars but not in polyenes. Through changing the shape of the polyene and substituting some hydrogen atoms on it by groups (Supplementary Fig. 5), we find that groups affect substantially the energy gap between the $S_2$ state and the doubly excited states $S_1$ and $1B_u^-$, whereas the gap between the $S_2$ and $S_y$ states is independent of any modification to the conjugated backbone (Supplementary Table 4). This implies that energies of the $S_1$ and $1B_u^-$ states, and thus the competition between the $S_2 \rightarrow S_1$ decay and the Car-to-Chl energy transfer might be tuned by modifying the groups attached to the conjugated backbone.

**Energy transfer from Cars to Chls.** Contribution of the dark state to the Car-to-Chl energy transfer is still an open question. It is recently proposed that the dark state-mediated energy transfer rate is of the same magnitude as the $S_2$-mediated one[6, 21]. This is in contradiction with previous work[4, 25, 26]. The nonadiabatic transition from the $S_2$ to $S_y$ states is always present for longer Cars according to discussions in the previous sections. The $S_y$ state must be involved in the Car-to-Chl energy transfer. We thus further investigate the energy transfer capability of the $S_y$ state, which is important for understanding the energy transfer mechanism in photosynthesis and resolving the controversies in this respect.

We calculate the energy transfer in the RG-B850 and RG-B800 pairs in *Rps. acidophila* as linked by *dashed lines* in Fig. 2b. Energy flow in these two kinds of pairs has been supposed to dominate the Car-to-Chl energy transfer[24]. Figure 7a shows the calculated absorption spectrum of the RG-B800 pairs, which is comparable to the experimental spectrum of LH2 complex in *Rps. acidophila* strain 10050[3, 6]. Positions of the $Q_x$ and $Q_y$ peaks deviate from the experimental ones by <0.1 eV. A charge-transfer state, with the electron excited from the Car to the Chl, appears between the $S_2$ and $Q_x$ states, which can result in the formation of Car radicals as detected in experiments[4]. The rate of energy transfer is evaluated via $k = (2\pi/\hbar)|V_{DA}|^2 J_{DA}$, where $V_{DA}$ and $J_{DA}$ are the electronic coupling strength and the spectral overlap between donor (D) and acceptor (A) transitions, respectively. $V_{DA}$ is calculated within the framework of many-body Green's function theory.

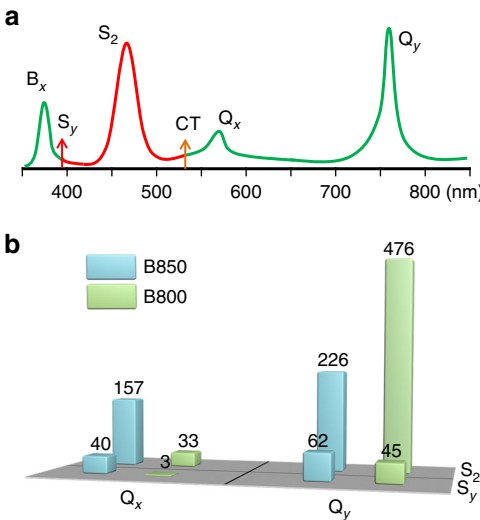

**Fig. 7** Absorption spectrum and electronic coupling strength for a carotenoid–chlorophyll complex. **a** Calculated absorption spectrum of the rhodopin glucoside-B800 complex as linked by the *dashed line* in Fig. 2b. CT denotes the charge-transfer state. Oscillator strengths of the $S_y$ and CT states are very weak. Here *arrows* are used to indicate their positions. **b** Electronic coupling strengths (in cm$^{-1}$) for the energy transfer passways from the $S_2$ and $S_y$ states to the $Q_x$ and $Q_y$ states in the rhodopin glucoside-B850 and rhodopin glucoside-B800 pairs as linked by *dashed lines* in Fig. 2b. Distances between the centers of corresponding rhodopin glucoside and B850/B800 molecules are given in Fig. 2b

We use the spectral overlap data from Krueger et al.[24] and Ostroumov et al.[21], where $J_{DA}(S_2–Q_x) = 13 J_{DA}(S_2–Q_y)$ and $J_{DA}(S_2–Q_x) = J_{DA}(S_y–Q_x)$. Figure 7b compares the electronic coupling strength for each energy transfer passway from Car to Chl. We find that the energy transfer rate of the $S_2$-B800 $Q_y$ channel is much higher than expected theoretically before[24] and can reach 70% of that of the $S_2$-B850 $Q_x$ channel which has been considered to be the dominating channel. This branching ratio is in good agreement with the experiment[39], supporting the experimental observations that both $Q_x$ and $Q_y$ have the role of energy acceptors and the amount of energy transferred to B800 is comparable to that to B850[4, 39–42]. The $S_y$-mediated energy transfer is predominated by the $S_y$-B850 $Q_x$ channel. The overall $S_y$-mediated energy transfer rate is one order of magnitude smaller than that of the $S_2$-mediated one. Thus, although the dark state participates in the energy transfer process, the Car-to-Chl energy transfer is still governed by the $S_2$ state (see Fig. 2c for the energy flow).

## Discussion

Ever since the discovery of the intermediate dark state between the $S_2$ and $S_1$ states in Cars by Cerullo et al. in 2002, its role in photosynthesis becomes increasingly emphasized[5]. On the basis of the conventional theoretical model for the electronic structures of polyenes developed by Tavan and Schulten[36], the $1B_u^-$ state has long been regarded as the candidate for the dark state. The strongly dependence of the $S_2$–$1B_u^-$ energy gap on the conjugation length make this assignment questionable[4, 9]. The $S_y$ state of the $A_g^+$ symmetry has a more moderate conjugation length dependence than the $1B_u^-$ state. More importantly, variation of the $S_y$ emission energy with respect to the conjugation length is parallel to the $S_2$ state, and their energy gap is kept at a small value (~ 0.1 eV) for Cars with $N > 9$. This is consistent with the experimental findings, such as those in 2DES, that the emission energy gap between the

dark state and the $S_2$ state is about 300 cm$^{-1}$ and independent of the conjugation length[4, 6, 10, 21]. This small energy gap ensures the extremely fast (~10 fs) internal conversion from the $S_2$ state to the dark state as observed experimentally[4, 5, 11]. The agreement between our theoretical 2D spectrum (Fig. 6) and the experimental data lends support to our model with a new $A_g^+$ state below $S_2$ at a large bond-length alternation. Interestingly, the experimental 2D study resolves the diagonal peak due to the dark state. Furthermore, based on ultrafast-transient absorption and transient-grating experiments, some groups have suggested that the dark-state is due to a double-minima structure on the $S_2$ potential energy surface, not a distinct electronic state[43, 44]. This double-minima model hypothesizes that the lowest one of the $S_2$ potential surface minima exists at the conformation where the carotenoid is twisted from the all-*trans* isomer by ~90° with respect to one C=C bond of the conjugated polyene backbone. This may be true for molecules with a short conjugated backbone such as the protonated Schiff bases[45], which is also demonstrated in our previous theoretical work[46]. Nevertheless, this does not seem to be the case for molecules with a long conjugated backbone (Supplementary Fig. 10). The new state described in the present article, with strong energy shift along the bond length alternation coordinate that crossovers in energy with the $S_2$ state (Fig. 3), does exhibit a double-well like feature that may explain the spectral shifts observed in the recent experiments. Noticeably, the theoretical 2D spectrum based on our model correctly describes the $S_2/S_y$ cross-peak above the diagonal in the experiment, whereas in the double-minima model one should expect to see extensive spectral diffusion and an elongated $S_2$ peak along the detection wavelength (below the diagonal), which is inconsistent with the 2D experimental data. In addition, the clear diagonal $S_y$ peak is also not explained by the double-minima model. On the basis of the above analysis, a carotenoid photophysics four-state model is given in Fig. 1c, which involves the $S_1$, $S_y$, and $S_2$ excited states.

One important motive to study the dark state in experiments is to solve the obvious distinction in the estimated Car-to-Chl energy transfer efficiency between previous theoretical calculations (20%) and experimental measurements (50–60%)[6, 21, 24]. Above we have illustrated that the portion of energy transferred via the $S_y$ state is minor based on the assumption $J_{DA}(S_2–Q_x) = J_{DA}(S_y–Q_x)$ proposed by Ostroumov et al.[21, 27]. Even if $J_{DA}(S_y–Q_x) > J_{DA}(S_2–Q_x)$, considering the smaller energy gap between the $S_y$ and $Q_x$ states than that between the $S_2$ and $Q_x$ states, the contribution of the $S_y$ state cannot fill the gap between theoretical calculations and experiments. The electronic coupling strengths we calculate by many-body Green's function theory are 1.5 times stronger than those from previous theoretical calculations by the transition density-cube approach[24]. We suggest that the disagreement in energy transfer efficiency between previous theoretical work and experiments may be due not only to the absence of the dark state in the theoretical model, but also to the underestimation of electronic coupling strengths in previous calculations.

In conclusion, our study provides evidence for a new state $S_y$ of the $A_g^+$ symmetry in Cars, thus contributing to a better understanding of carotenoid excited states. Future experiments would be required to test the accuracy of our calculations.

## Methods

**Ground-state geometry.** Density-functional theory (DFT) with the Coulomb-attenuating method variant of the Becke 3-parameter-Lee-Yang-Parr (CAM-B3LYP) exchange-correlation functional[47] is used to optimize geometries of Cars and Chls by the Gaussian 09 program[48]. CAM-B3LYP has been shown to give more reasonable structures than other functionals[14, 49].

**Excitation energy.** Many-body Green's function theory, which includes the combination of GW method and Bethe–Salpeter equation (BSE)[50, 51], is applied

to compute the excitation energies with a Gaussian orbital based GW-BSE package[52, 53]. Calculations are performed at the level of full BSE, i.e., considering the mixing between resonant and antiresonant transitions, as Tamm–Dancoff approximation can cause large errors for organic molecules[54–56]. This scheme has been applied for electronic excitations in many organic systems[54, 57–59]. The $S_1$ and $1B_u^-$ states bear doubly excited character, involving two coupled triplet excitations. Their excitation energies cannot be obtained from BSE directly as BSE can only deal with one electron–hole pair excitation. Here, we estimate their energies according to Tavan and Schulten's theory[36], i.e., $E(S_1) = 2E(T_1)$ and $E(1B_u^-) = E(T_1) + E(T_2)$, where $T_1$ and $T_2$ are the lowest two triplet states of Cars and are computed via GW-BSE. This approach possesses high accuracy as proved by Tavan and Schulten. In the Supplementary Note 1, Supplementary Fig. 1, and Supplementary Table 1, a detailed discussion on the accuracy of our strategy to predict the $S_1$ and $1B_u^-$ energies is presented.

**Potential minimum at the excited state**. Equilibrium structure of Cars in the $S_2$ state, which originates from the HOMO → LUMO transition (point $R_2$ in Fig. 3) is optimized by the constrained density-functional theory (CDFT) where occupations in HOMO and LUMO are fixed at 1 during structural relaxation[60]. As the $S_y$ state is composed predominantly by transitions HOMO-1 → LUMO and HOMO → LUMO+1 with equal weight, its potential minimum cannot be predicted by structural optimization on its own energy surface using CDFT. We locate the potential minimum of the $S_y$ state (point $R_y$ in Fig. 3) approximately by extrapolating along the $R_0 → R_2$ reaction coordinate of the $S_2$ state. We validate the reasonability of the $S_y$ potential minimum predicted by this scheme through relaxing Cars in the $S_y$ state with excited-state forces from BSE. BSE forces are computed by the approach proposed by Ismail-Beigi and Louie with the exception that we use finite difference method to evaluate derivatives of single-particle energies and wave functions with respect to nuclear positions[61, 62]. This technique to compute BSE forces has been well tested to give consistent results with those from Ismail-Beigi and Louie on the excited-state structures of CO and $NH_3$ molecules[61] (see Supplementary Note 3, Supplementary Table 5, and Supplementary Figs. 6 and 7 for details of the theory and test of the accuracy).

**Theoretical 2D spectrum**. We utilize a density matrix-based dynamical method to simulate 2D spectrum of *Rps. acidophila*. This method accounts for full density-matrix dynamics and bath memory effects in the simulated 2D spectrum, and details of the theory are described in refs [63, 64]. The model adopted for the *Rps. acidophila* system consists of five carotenoid states ($S_0$, $S_1$, $S_y$, $S_2$, and $S_n$) and one chlorophyll state ($Q_x$). The transition energies are set at 16,900, 17,810, 18,620, and 17,500 cm$^{-1}$ for $Q_x$, $S_y$, $S_2$, and $S_1 → S_n$, respectively. This model places the $S_y$ energy at 0.1 eV below the $S_2$ state to describe carotenoid electronic states at a geometry highly displaced along the bond length-alternation coordinate, in accordance with our calculations for RG. The dynamics are described by a Lindblad master equation including five population relaxation terms: $\tau_{S_2→S_y} = 0.3$ ps, $\tau_{S_y→S_1} = 1$ ps, $\tau_{S_y→Q_x} = 2$ ps, $\tau_{S_1→S_0} = 3$ ps, $\tau_{Q_x→S_0} = 1$ ps. The line broadenings are described by Gaussian static disorders with $\sigma = 450$ cm$^{-1}$, and couplings to a super-Ohmic bath with a spectral density $J(\omega) = \gamma_0 \frac{\omega^3}{\omega_c^2} e^{-\omega/\omega_c}$. The coupling strength $\gamma_0$ and cutoff $\omega_c$ are set at 0.6 and 850 cm$^{-1}$, respectively. Note that in this work, we aim to demonstrate that our model produces 2D spectrum that is consistent with experimental results, therefore we do not perform full fittings to the experimental spectra, and the bath and dynamical parameters used in our model are only set tentatively. Additional calculations that provide nonadiabatic couplings and explore the potential surfaces more completely are required to fully describe the experimental 2D data.

**Energy transfer**. The electronic energy transfer rate, $k$, is calculated via $k = (2\pi/\hbar)|V_{DA}|^2 J_{DA}$ according to the Förster–Dexter theory. The electronic coupling strength $V_{DA}$ is computed within the framework of many-body Green's function theory via

$$V_{DA} = X_D^T \left( K_{DA}^x + K_{DA}^d \right) X_A - \omega_0 X_D^T S_{DA} X_A \qquad (1)$$

where $X_{D/A}$ is the BSE exciton wave function for donor (D) and acceptor (A), $\omega_0$ the energy transferred, $S_{DA}$ the overlap matrix between donor and acceptor orbitals. $K^x$ and $K^d$ are the exchange and direct terms of the BSE electron–hole interaction kernel. Electronic coupling arising from these two terms resemble the Förster and Dexter coupling, respectively[65]. $V_{DA}$ computed by Eq. (1) consists excellently with that by the high-level quantum chemistry approach CASSCF[66].

**Data availability**. All data supporting the findings of this study are available from the corresponding author on request.

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

## Acknowledgements

This work was supported by the National Natural Science Foundation of China (Grants Nos. 21433006, 21573131, 21173130, and 21603056) and the Natural Science Foundation of Shandong Province (Grant No. JQ201603). Y.-C.C. thanks the Ministry of Science and Technology, Taiwan (Grant No. NSC 105-2113-M-002-012), National Taiwan University (Grant No. 103R891305), and Center for Quantum Science and Engineering (Subproject: 103R891401) for financial support. Computational resources have been provided by the National Supercomputing Centers in Jinan.

## Author contributions

Y.M. designed the calculations and J.F. carried out most of the calculations. C.-W.T. and Y.-C.C. performed the calculation of 2D spectra. T.C., X.L., and H.Y. took part in the optimization of configurations. Y.M. and M.R. performed coding on the many-body Green's function theory and Förster–Dexter theory. Y.M., J.F., and Y.-C.C. wrote the manuscript.

## Additional information

**Competing interests:** The authors declare no competing financial interests.

