## [Peer Review File · Nature Communications]

Reviewers' comments:

Reviewer #1 (Remarks to the Author):

The present manuscript is reporting a new proposal for the mechanism of energy transfer from carotenoids to chlorophylls. The manuscript is clearly written and the implications of the newly proposed hypothesis are well explained and discussed. Although this positive impression, I am rather sceptical about some technical weakness of the proposed calculations. The model is based on the accurate calculation of (small) energy differences between several excited states calculated on ground and excited state optimised geometries. These energetics is actually evaluated and validated using a mixture of several approximations, and it is not clear to me which is the uncertainty that should be attributed to the final results.

I believe that the aim of computational chemistry should be to provide these energies with an accuracy such that the picture of the energetics of the different levels can be quantitatively drawn.

I think that for several reasons this is not the case:

- The approximation used for the calculations of the excited state gradient is rather poor and it is not clear how much is the error committed by using this approximation.
- Another important point is concerning the calculations of evaluation of the excited state energies with double excitation character. The whole model is based on energy differences between single and double excitations, and it is therefore important to have an unbiased estimate of both. As clearly written in the methods section the authors cannot provide a direct measurements of double excitation energies a they use an approximation which may provide biased results.
- If the goal of the manuscript is to compare with electron transfer occurring in photosynthetic proteins the authors should (quantitatively and explicitly) consider the effect of the protein environment in their calculations.

In summary, I am rather convinced that the interesting picture of the carotenoid photophysics described by the authors can be correct, but this hypothesis is not fully and quantitatively supported by the reported data.

Reviewer #2 (Remarks to the Author):

The authors present an attractive hypothesis modifying the current knowledge about excited states of carotenoids, yet I believe the proposed scheme of excited states remains hypothetical and thus is not suitable for publication in Nat. Comm. Adding another excited state into a plethora of various dark states present in the carotenoid excited state manifold needs more solid proofs and arguments. My reasons for this conclusions are the following:

1. The first sentence in abstract says that "...it has been generally accepted that a dark excited state in carotenoids plays a crucial role in EET...". This claim is in fact correct, but it refers to the dark S1 state that is indeed generally accepted as the important energy donor in light-harvesting systems. However, it is certainly not true if we talk about the Bu- state, which has been suggested as a potential donor only a few times, and it is far from "being generally accepted".

2. In Conclusion paragraph starting in line 273 the authors argue that they "...propose a new model of carotenoid photophysics...". This is clearly an overstatement, because, looking at Fig. 1, the model is no different from the "Bu-" model, except the role of the Bu- state now plays the hypothetical "Sy" state.

3. I am missing detailed explanation why this "Sy" state has not been observed in other quantum chemical calculations. Besides the methods the authors refer to in lines 96-97, there exist other approaches as for example high-level semiempirical calculations (SAC-CI, EOM-CCSD) from Birge's

group.

4. Another issue is the missing symmetry label of the S_y state. Why it is not there? What prevents the authors to assign a symmetry label to this state? For example, can the authors rule out the possibility that the S_y state is just another potential minimum of the S_2 (Bu^+) state? Possibility of multiple minima on the S_2 (and S_1) potential surfaces was explored by some authors (e.g. de Weerd CPL 2002, Ghosh JPCB 2015). This possibility is not explored at all in the manuscript and the authors simply assume that each state has only a single potential minimum.

5. Line 103: the magnitude of the solution-induced red-shift depends on the solvent thus it can hardly be concluded that it is 0.1 eV, making the agreement between calculations and experiment excellent. It will be less than 0.1 eV for hexane and more than 0.1 eV for highly-polarizable solvents such as CS_2 .

6. Line 109: it is dangerous to compare excited state energies of RG and b-carotene. Although they have formally the same number of conjugated $C=C$ bonds, effective conjugation length of b-carotene is shorter than of RG due to extension of conjugation to terminal rings in b-carotene (see e.g. Fuciman et al. JPCB 2015)

7. In line 113 the authors argue that the new state is a singly-excited state, yet it is a combination of two transitions. I believe these two claims contradict each other.

8. In their arguments, the authors obviously mix two different states discussed in literature. While they try solve the " Bu^- problem", references 8 and 25 deals with another state, tentatively denoted as the S^* state. I admit that in the first studies of the S^* state it was speculated that it might be the Bu^- state, but later it was clearly shown that S^* state (Refs 8 and 25) has nothing common with the Bu^- state.

9. Line 208: could the authors provide a reference to "experimental RG-B800 spectrum"? I do not recall that such spectrum has ever been measured experimentally.

10. The excitation and emission energy of the Bu^- state is quite complicated issue that is far from being settled. The " Bu^- emission" was reported only by Koyama group (Ref 19, not 18 as incorrectly stated in line 65) and the energies extracted from data in Fig. 19 are based solely on fitting and it is not clear whether the data contains any Bu^- emission at all. Thus, I would be careful to claim that Bu^- emission energy is less than 2 eV.

Reviewer #3 (Remarks to the Author):

This paper addresses a long-standing topical issue regarding the electronic states of carotenoids and their roles in light harvesting. The main advance of the paper is the improved electronic structure calculations compared to previous studies. The conclusion of the paper suggests that this work is the last word on the subject, which I think is misleading and overstates the advance and its generality. Nevertheless, the work is clear and is a significant result that is well suited for publication in this journal.

The precise nature and assignment of S_y is not completely clear to me. Is this likely to be the " x " state that was observed by Ostroumov et al in their 2D spectroscopic studies of LH2?

Reviewer #1:

Comments: *The present manuscript is reporting a new proposal for the mechanism of energy transfer from carotenoids to chlorophylls. The manuscript is clearly written and the implications of the newly proposed hypothesis are well explained and discussed. Although this positive impression, I am rather sceptical about some technical weakness of the proposed calculations. The model is based on the accurate calculation of (small) energy differences between several excited states calculated on ground and excited state optimised geometries. These energetics is actually evaluated and validated using a mixture of several approximations, and it is not clear to me which is the uncertainty that should be attributed to the final results.*

I believe that the aim of computational chemistry should be to provide these energies with an accuracy such that the picture of the energetics of the different levels can be quantitatively drawn. I think that for several reasons this is not the case:

Reply: We would like to thank the reviewer to raise this important question concerning the accuracy on the excited-state dynamics. We fully agree with the reviewer that enough accuracy is required to get reliable picture on the energetics of different excited-state levels. Different from studies of ground-state properties whose computational cost is relatively low and analytic energy gradient is available easily, calculations on the excited state, especially excited-state geometry optimization, are not easy tasks from the points of both accuracy and efficiency. High-level quantum chemistry methods, such as CCSD, CASPT2 and quantum Monte Carlo, are good at the accuracy, but their computation is very demanding for large systems such as the carotenoids studied in our work. Many-body Green's function theory (GW-BSE) is an excited-state theory for single electron excitation and has been demonstrated to reach the accuracy comparable to these superior quantum chemistry methods. It has predicted successfully excited-state properties of both solids and molecules. However, analytic formula for the excited-state force within GW-BSE are not available yet. Thus we have to apply other effective approaches, such as constrained DFT (CDFT) and TDDFT, to optimize the geometry in the excited state, after which GW-BSE is utilized to get more accurate excitation energies. In CDFT and TDDFT the excited-state force can be calculated analytically. Although the excitation energy from CDFT and TDDFT is not accurate enough, the profile of the energy surface, the trend for the system to relax in the excited state and position of the excited-state minimum may be reasonably computed. Although the accuracy is not very perfect quantitatively as we wish, the final conclusion and picture is reasonable

and right qualitatively. This can be seen from the following replies to other comments and from the discussion added to the Supplementary of the revised version of the manuscript. Besides the excited-state dynamics results obtained from CDFT that is discussed in the main context, we also present some results based on the TDDFT optimized excited-state geometries using two kinds of exchange-correlation functionals in the Supplementary of the revised manuscript. Although there is some difference in the exact energies for the vertical excitation and Stokes shift of the S_2 and S_y states in these three schemes (CDFT and two TDDFT), our key conclusions, i.e. S_2 can decay into S_y and the emission of S_y is lower than that of S_2 , are identical. These conclusions are also supported by our other calculations which are also provided in the revised manuscript and the Supplementary. For example, (i), S_y is a new A_g^+ state and remains optically forbidden in both the all-*trans* and *cis* configurations, explaining why it is difficult to be detected in experiments. (ii), the simulated two-dimensional electronic spectra for the carotenoid-chlorophyll energy transfer based on our results are consistent with the experiment. Therefore, we think that our results are solid.

1. **Comments:** *The approximation used for the calculations of the excited state gradient is rather poor and it is not clear how much is the error committed by using this approximation.*

Reply: As discussed above there is no analytic formula for the BSE excited-state force available yet. It is quite time-consuming to compute BSE excited-state force by the finite difference approach directly which requires $2*(3N-6)$ GW-BSE runs for each structure optimization step if the second-order central difference approximation for the derivative is utilized. Here, N is the number of atoms in the system and $3N-6$ is the number of degrees of freedom. In 2003 Ismail-Beigi and Louie proposed a scheme to evaluate BSE force approximately [Phys. Rev. Lett. 90, 076401 (2003)]. The main computational load of this scheme is to acquire the derivative of ground state electronic levels $\partial_R E_i$ and wave functions $\partial_R |i\rangle$ with respect to nuclear position R . $\partial_R E_i$ and $\partial_R |i\rangle$ are used to constitute the BSE force in their scheme. Ismail-Beigi and Louie employ density functional perturbation theory and first order perturbation theory to obtain $\partial_R E_i$ and $\partial_R |i\rangle$. Following their idea we develop our own code to compute BSE force, but we use the simple finite difference approach to get $\partial_R E_i$ and $\partial_R |i\rangle$. We obtain similar excited-state structures and energies for CO and NH_3 molecules to the work by Ismail-Beigi and Louie, the results from other quantum chemistry approaches such as EOM-CCSD and CASSCF and the experimental data. Details of our approaches and its test on CO and NH_3 are provided in the Supplementary Section 5 of the revised manuscript.

When applying our BSE force to relax the carotenoid in the S_2 state, we find that the result is not good and a reasonable equilibrium structure cannot be obtained. This is mostly because the energy surface of the S_2 state is too flat, the total energy of the S_2 state lowers just by 0.1 eV (see Supplementary Figure S7 and S8). The error in our approximate BSE force for the S_2 state may be of the same magnitude as the exact force, which makes the minimization procedure, by the conjugate-gradient technique in our work, hard to converge to the right structure. For the CO and NH_3 molecules together with the S_y state for the carotenoid, our BSE force scheme can get reasonable excited-state structures since the energy surfaces around these excited-state minima are steep, for example the total energy of the S_y state drops by 0.6 eV during relaxation starting from the ground-state geometry (see Supplementary Figure S7 and S8). Based on these points we employ another strategy to relax carotenoids in the S_2 state.

In the work by Ismail-Beigi and Louie for CO and NH_3 molecules in Phys. Rev. Lett. 90, 076401 (2003) and by Artacho *et al.* for polymer in Phys. Rev. Lett. 93, 116401 (2004), CDFT is demonstrated to give reliable excited-state structures and the accuracy can even be comparable to EOM-CCSD and CASSCF. The combination of CDFT and GW-BSE is shown in the latter literature to be an effective technique to evaluate energy shift in the excited-state relaxation. The S_2 state of carotenoid is predominantly composed by the HOMO \rightarrow LUMO transition, which makes CDFT applicable to it. Based on the CDFT-optimized excited-state structure of S_2 (configuration R_2 in Fig. 3), we calculate the Stokes shift of the S_2 state to be 0.18 eV for rhodopin glucoside and spirilloxanthin, close to the experimental measurement. Energy of the S_y state decreases much faster and becomes nearly degenerate with the S_2 state at R_2 . To prove the validity of our conclusions based on the CDFT+GW+BSE technique, we further apply the TDDFT+GW+BSE approach where TDDFT is used to get the S_2 state minimum. The results are presented in Supplementary Figure S7 and S8 of the revised manuscript. TDDFT is a well-known effective approach to perform excited-state optimization where efficient analytic excited-state force formula such as the Z-vector equation [J. Chem. Phys. 117, 7433 (2002)] have been used widely. Two functionals, CAM-B3LYP and B3LYP, are utilized to exam the dependence of results on methods. As shown in Supplementary Figure S7 and S8, calculations based on the TDDFT/CAM-B3LYP and TDDFT/B3LYP optimized S_2 geometries come to the same conclusions as CDFT that S_2 can decay into S_y and the emission of S_y is lower than that of S_2 . Thus, our conclusions are robust against the difference in the technique to relax carotenoids in the excited state. The error in the excited-state gradient is not so important for our conclusions. Although our technique to calculate the excited-state force and energy surface is rough, it can still predict the trend of excited-state relaxation and locate the

minimum of the S_2 state reasonably. From Supplementary Figure S7 and S8, the energy difference between the S_2 and S_y states at the R_2 configuration is little, at the same order as or even smaller than the vibration energy of the C-C and C=C bonds (the stretching modes of C-C and C=C are 1195 and 1590 cm^{-1} , respectively). Some errors in our calculations do not influence our conclusions since the conjugated backbone vibration, which is not considered in our work, can also play a role in the $S_2 \rightarrow S_y$ transition.

Using the above CDFT+GW+BSE and TDDFT+GW+BSE techniques we could predict the emission energy of the S_y state to be 0.1~0.2 eV smaller than the S_2 state after extrapolating further along the $R_0 \rightarrow R_2$ reaction coordinate. This is done based on the S_2 energy surface calculation. One question that has to be answered before making the final conclusions is that whether the S_y state will fall by a large amount ($\gg 0.2$ eV) after passing R_2 on its own surface or not. If it is true, the gap between the emission energies of the S_2 and S_y states may be too large to account for the experiments, because the gap between the emission energies of S_2 and the dark states is ~0.1 eV according to the two-dimensional electronic spectra in Science 340, 52-56 (2013), J. Chem. Phys. 142, 212433 (2015) and the high time resolution broadband pump-probe spectroscopy in Phys. Chem. Chem. Phys. 14, 6312-6319 (2012). To check this issue, we relax the carotenoid in the S_y state by our BSE excited-state force approach. This work cannot be done by CDFT since the composition of S_y is complicated, both HOMO \rightarrow LUMO+1 and HOMO-1 \rightarrow LUMO transitions having important contribution. Modifying the occupation in these orbitals simply, which is what CDFT does, cannot simulate the S_y state. Since the potential well around the S_y minimum is steep as discussed above, our BSE excited-state force approach can give a good prediction of the S_y minimum. Our studies show that the emission energy of S_y calculated by the BSE force is close to that got by the extrapolation approach along the $R_0 \rightarrow R_2$ reaction coordinate, which proves that the emission energy of S_y does be a little lower than that of S_2 .

2. **Comments:** *Another important point is concerning the calculations of evaluation of the excited state energies with double excitation character. The whole model is based on energy differences between single and double excitations, and it is therefore important to have an unbiased estimate of both. As clearly written in the methods section the authors cannot provide a direct measurements of double excitation energies a they use an approximation which may provide biased results.*

Reply: The doubly excited states of carotenoids usually refer to the states like $2A_g^-$, $1B_u^-$, etc. which are formed by two coupled triplet excitations. To reach these states from the ground state, two electrons are excited with their spin

flipping to create two triplets. For example, $2A_g^-$ is constituted by two T_1 , $1B_u^-$ by T_1+T_2 , where T_1 and T_2 are the lowest and the second lowest triplet excitons. The single-electron excitation theory cannot simulate these doubly excited states. BSE is a theory to model single-electron excitation.

Our work in this manuscript focuses on the S_2 and S_y states and their energy difference. Both S_2 and S_y states are singly excited states and bear no double excitation character. It is easily to understand that S_2 is a singly excited state since it is formed predominantly by the HOMO \rightarrow LUMO transition. We are sorry that we did not explain clearly in the old manuscript that S_y is a singly excited state too. Now Supplementary Section 3 of the revised manuscript discusses in detail the composition and wave function of S_y and other excited states. S_y is composed dominantly by the HOMO \rightarrow LUMO+1 and HOMO-1 \rightarrow LUMO transitions. These two components do not mean that one electron is promoted from HOMO to LUMO+1 and another from HOMO-1 to LUMO, and do not mean that S_y is a double excitation. It only means that the wave function of the S_y exciton can be represented by the linear combination of $\psi_{H-1}(\mathbf{r}_h)\psi_L(\mathbf{r}_e)$ and $\psi_H(\mathbf{r}_h)\psi_{L+1}(\mathbf{r}_e)$ in the language of BSE, where \mathbf{r}_h and \mathbf{r}_e are coordinates of the hole and excited electron respectively (see Supplementary Section 3 for details), and spatial distribution of the hole (the excited electron) can be described by the combination of HOMO-1 and HOMO (LUMO and LUMO+1). The other quantum chemistry method EOM-CCSD also finds this state with similar composition (see Supplementary Section 3 for details). In EOM-CCSD there exists some double excitation contribution from HOMO² \rightarrow LUMO² to S_y . However the weight of this double excitation component is too weak (one order of magnitude smaller than those from HOMO \rightarrow LUMO+1 and HOMO-1 \rightarrow LUMO) to change the single excitation feature of S_y . Above S_y there is another state which is also composed by HOMO \rightarrow LUMO+1 and HOMO-1 \rightarrow LUMO but with different phase from S_y . This state can be attributed to the singly excited $1A_g^+$ state which has been well defined in the community (see Supplementary Section 3 for details).

Since S_2 and S_y are both singly excited states and BSE is a theory for single-electron excitation, our calculations for S_2 and S_y are on the even ground without bias to any one of them. Thus the calculated energy difference between S_2 and S_y is of high accuracy and trustable.

As described above, S_y has nothing to do with doubly excited state. However, to test the accuracy of our GW-BSE methods from another aspect, we study the excitation energy of the doubly excited states $2A_g^-$ and $1B_u^-$. These two states are formed by two triplets, involving excitation of two electrons. They are beyond the ability of BSE. Based on their calculations using the multiple-reference double

excitation configuration-interaction method (MRD-CI), Tavan and Schulten [Phys. Rev. B 36, 4337 (1987)] find that energies of the $2A_g^-$ and $1B_u^-$ states of polyenes can be estimated by $E(2A_g^-) \approx 2E(T_1)$ and $E(1B_u^-) \approx E(T_1) + E(T_2)$, where T_1 and T_2 are the lowest two triplet states of Cars. Accuracy of this approximation for the absolute energies of these two states is 0.1 eV. In Supplementary Section 1 of the revised manuscript, we present our test of this approach on β -carotene whose results from other quantum chemistry approach DFT/MRCI are available. Calculation shows that accuracy of the energy difference between $1B_u^-$ and S_2 by this approach based on BSE is high. Therefore, our energies for the $2A_g^-$ and $1B_u^-$ states of rhodopin glucoside and spirilloxanthin are reliable. Just as the reviewer comments, our model is based on the energy difference between the S_2 and S_y states. These two states are both singly excited ones as we have explained above, but not S_2 the singly one while S_y the doubly one as the reviewer understood (we are sorry that we did not explain clearly in the old manuscript which leads to this misunderstanding). Therefore, the approximations in $2A_g^-$ and $1B_u^-$ do not affect our model and conclusion.

3. **Comments:** *If the goal of the manuscript is to compare with electron transfer occurring in photosynthetic proteins the authors should (quantitatively and explicitly) consider the effect of the protein environment in their calculations.*

Reply: We would like to thank the reviewer to remind us the effect of the protein environment. In fact, our manuscript talks about the energy transfer between carotenoid and chlorophyll in the photosynthetic process, not the electron transfer behavior. The energy transfer process is as the following: first, carotenoid absorbs a photon and is promoted to the excited state; second, carotenoid relaxes in the excited state to its potential minimum; finally, carotenoid decays radiatively to the ground state and emits a photon which is absorbed by the chlorophyll simultaneously. Energy transfer is realized through fluorescence resonance, not by electron transfer. According to the suggestion of the reviewer, we exam the effect of protein environment on the excitation energies of carotenoid. The results are given in Supplementary Section 2 in the revised manuscript and discussed on Page 5 in the main context. Three protein fragments are chosen to study the influence of fragment size and position on the excitation energies. Results show that the effect of protein fragments on S_2 and S_y is limited and does not alter the conclusion of our work.

4. **Comments:** *In summary, I am rather convinced that the interesting picture of the carotenoid photophysics described by the authors can be correct, but this hypothesis is not fully and quantitatively supported by the reported data.*

Reply: We appreciate that the reviewer believes our results. We supplement more materials, as listed in the Main changes to the manuscript on the first page of our replies to the reviewers' comments, in the revised manuscript according to the reviewer's comments and suggestions. More theoretical approaches are employed to validate our conclusions and our results are supported by more experimental evidences. We think that our hypothesis can now be fully and quantitatively supported by the reported data.

Reviewer #2:

Comments: *The authors present an attractive hypothesis modifying the current knowledge about excited states of carotenoids, yet I believe the proposed scheme of excited states remains hypothetical and thus is not suitable for publication in Nat. Comm. Adding another excited state into a plethora of various dark states present in the carotenoid excited state manifold needs more solid proofs and arguments. My reasons for this conclusions are the following:*

Reply: We would like to thank the reviewer for the valuable comments and suggestions that help us exam more deeply the new state we find and improve the manuscript greatly. After performing additional calculations and studies according to the comments of the reviewers, we are more confident that the S_y state we find is a new A_g^+ state which can also be obtained from other high-level quantum chemistry method EOM-CCSD. In the revised manuscript and the corresponding Supplementary, we describe and discuss in more detail the characters of this state including the symmetry, wave function, variation of energy and oscillator strength with the twist of geometry, etc. Through comparison with the experiments and simulating the two-dimensional electronic spectrum, the S_y state does take part in the relaxation process of the S_2 state and the energy transfer between carotenoid and chlorophyll. We think that we have provided solid proofs and arguments to support our conclusions.

1. **Comments:** *The first sentence in abstract says that "...it has been generally accepted that a dark excited state in carotenoids plays a crucial role in EET...". This claim is in fact correct, but it refers to the dark S1 state that is indeed generally accepted as the important energy donor in light-harvesting systems. However, it is certainly not true if we talk about the Bu- state, which has been suggested as a potential donor only a few times, and it is far from "being generally accepted".*

Reply: We appreciate that the reviewer points out this flaw in the expression of first sentence. In the sentence "...it has been generally accepted that a dark excited state in carotenoids plays a crucial role in EET...", the dark state does not refer to S_1 but an intermediate state between the S_1 and S_2 states. We are sorry for this unclarity. This dark state also does not refer to $1B_u^-$ since its role as the intermediate state is questioned and far from being generally accepted just like the reviewer says. The nature of the dark state is still unclear. This is the reason why we study it in this work. To be more rigorous, we remove this sentence and substitute it by "**It has long been controversial whether there is an intermediate dark state between the S_2 and S_1 states of carotenoids. Recent two-dimensional electronic spectroscopy (2DES) measurements further substantiate its existence and its involvement in the energy transfer (ET) from carotenoids to chlorophylls.**". We use the word "controversial" here since although the majority of studies support the existence of the intermediate state between the S_1 and S_2 states, some experiments do not detect this state. Based on our calculations and the experiment in Phys. Chem. Chem. Phys. 14, 6312-6319 (2012), the intermediate state is very close in energy to the S_2 state (within 0.1 eV), and whether it is below or above the S_2 state is sensitive to the polarizability of environment. This might be the reason why some experiments do not observe the decay of S_2 to the intermediate state.

2. **Comments:** *In Conclusion paragraph starting in line 273 the authors argue that they "...propose a new model of carotenoid photophysics...". This is clearly an overstatement, because, looking at Fig. 1, the model is no different from the "Bu-" model, except the role of the Bu- state now plays the hypothetical "Sy" state.*

Reply: Since S_y is a new state, we thought that our model in Fig. 1c is a new one. To avoid ambiguity, we remove the sentence "...propose a new model of carotenoid photophysics..." and replace it with "**Based on above analysis, a carotenoid photophysics four-state model is given in Fig. 1c which involves the S_1 , S_y and S_2 excited states.**" in Page 11 according to the comment of the reviewer.

3. **Comments:** *I am missing detailed explanation why this "Sy" state has not been observed in other quantum chemical calculations. Besides the methods the authors refer to in lines 96-97, there exist other approaches as for example high-level semiempirical calculations (SAC-CI, EOM-CCSD) from Birge's group.*

Reply: We would like to thank the reviewer to remind us the theoretical work by Prof. Birge which we missed in the old manuscript. In the revised manuscript, some related publications from Birge's group are cited in the first paragraph of Results section (lines 96-97 of the old manuscript). We also calculate the excited states of the N=11 polyene by the EOM-CCSD method (Supplementary Section

3). We choose the N=11 polyene since the computational cost of EOM-CCSD for it is much lower than that for rhodopin glucoside (N=11) and excitation energies of the S_2 and S_y states which we concern for these two species are very close according to our investigation (see Supplementary Section 4). As discussed in Supplementary Section 3, EOM-CCSD can also get the S_y state. The energy gap between the S_2 and S_y states from EOM-CCSD is similar to that from GW-BSE. In the main context, some discussions on the EOM-CCSD results are added on Page 5 in the revised manuscript.

The S_y state may have already been calculated out by other quantum chemistry calculations. These calculations may focus on the known states, e.g. $2A_g^-$, $1B_u^-$, $1B_u^+$ and $1A_g^+$, and pay little attention to other states. For example, in the work on rhodopin glucoside by Magdaong *et al.* in J. Phys. Chem. B 118, 11172 (2014), several states appear between the S_2 and $1A_g^+$ states in their calculations (see Figure 13 in this reference). In the previous EOM-CCSD work on peridinin by Coccia *et al.* in J. Chem. Theory Comput. 10, 501 (2014), a state of similar composition (HOMO→LUMO+1 and HOMO-1→LUMO) and symmetry (A_g) to the S_y state is found to lie 0.4~0.6 eV above the S_2 state. Till now, no one has ever studied systematically the S_y state and its excited-state dynamics which is important to understand the decay from the S_2 to the S_y state. Our work is the first on this respect.

4. **Comments:** *Another issue is the missing symmetry label of the S_y state. Why it is not there? What prevents the authors to assign a symmetry label to this state? For example, can the authors rule out the possibility that the S_y state is just another potential minimum of the S_2 (Bu^+) state? Possibility of multiple minima on the S_2 (and S_1) potential surfaces was explored by some authors (e.g. de Weerd CPL 2002, Ghosh JPCB 2015). This possibility is not explored at all in the manuscript and the authors simply assume that each state has only a single potential minimum.*

Reply: We would like to thank the reviewer for these useful suggestions to help us characterize the S_y state more completely. According to its composition and its single-electron excitation character, the S_y state can be assigned to the A_g^+ symmetry. This point is stated on Page 5 in the revised manuscript. S_y is different from the $1A_g^+$ state which has been well defined in the community. Supplementary Section 3 discusses detailedly the disparity between S_y and $1A_g^+$. For example, (i) $1A_g^+$ is higher than S_y in energy; the energy gap between S_2 and $1A_g^+$ from our GW-BSE and EOM-CCSD calculations agree with experiments. (ii) Experiments find $1A_g^+$ changes from optically forbidden to allowed when the configuration of carotenoid changes from all-*trans* to *cis*; our calculations

reproduce this phenomenon; however, S_y remains optically forbidden in all the configurations, which might be the reason why S_y is hard to discern in the experimental spectra.

The S_y state is definitely not another potential minimum of the S_2 state since compositions of these two states are quite different. S_2 is formed mainly by HOMO→LUMO and S_y mainly by HOMO→LUMO+1 and HOMO-1→LUMO at both the ground-state geometry (R_0 in Fig. 3 of the manuscript) and the excited-state geometry (R_2 and R_y in Fig. 3 of the manuscript). Energy surface of the S_2 state intersects with that of the S_y state (see Fig. 3 of the manuscript, Supplementary Figure S7 and S8), leading to nonadiabatic transition from the S_2 to the S_y state.

The articles de Weerd CPL 2002 [Chem. Phys. Lett. 354, 38 (2002)] and Ghosh JPCB 2015 [J. Phys. Chem. B 119, 14905 (2015)] propose a hypothesis to explain the experiments. In their models, they assume that there are two minima on the S_2 potential surface. Ghosh suggests that the intermediate state below the S_2 state observed in the experiments, such as the state X in the two-dimensional electronic spectrum (2DES) in Science 340, 52 (2013), is the lower minimum of the S_2 state, while the S_2 peak in the diagonal of the 2DES originates from the S_2 emission at a higher minimum. If the X state is one of the minima of the S_2 state just like Ghosh suggests, there will be no diagonal X peak and S_2/S_y cross-peak above the diagonal as appear in the experimental 2DES, on the contrary some extensive spectral diffusion and an elongated S_2 peak along the detection wavelength (below the diagonal) are expected, which is inconsistent with the 2D experimental data. Detailed discussion on the 2DES, together with our new work on simulated 2DES, is given on Page 7, 8 and 11 in the revised manuscript.

In their models, de Weerd and Ghosh hypothesize that the first (higher) minimum of S_2 comes from bond-length alternation while the second (lower) minimum of S_2 is reached by twisting around one of the C=C bonds of the conjugated polyene backbone. They assume that long carotenoids have the same potential surface profile with respect to the torsional coordinate as the protonated Schiff bases with $N=5$. In the 3-state model for the protonated Schiff bases, it does have the lowest minimum for the first bright state at the torsional angle around 90° [see Fig. 1 in Phys. Rev. Lett. 112, 238301 (2014) and Fig. 25 in Ghosh JPCB 2015]. In our previous GW-BSE calculations on the protonated Schiff bases [Phys. Rev. B 81, 115433 (2010)], we prove the 3-state model and locate the lowest minimum of the first bright state at the torsional angle around 90° . However, one of the problems of this model for the carotenoids studied in our work (most possibly for other carotenoids too) is that the emission energy at

the lowest minimum is too small to account for the X state in the 2DES [emission energy of the X state is just ~ 0.1 eV smaller than that of the S_2 state in Science 340, 52 (2013)].

To make clear what is going on for carotenoids which we study, we explore the potential surface of the N=11 polyene with respect to the torsional coordinate around one of the C=C bonds of the conjugated backbone (see Supplementary Section 7) according to the suggestion of the reviewer. We study two cases, (i) rotating around C(11)=C(11') and (ii) rotating around C(5)=C(6) from the all-*trans* to *cis* configurations. Please note that since we do not take the bond-length alternation into account, there is no local minimum in the vicinity of the all-*trans* configuration. For both cases, S_2 reaches a potential maximum but not a minimum at the torsional angle of 90° . The excitation energy at this maximum is 0.7 and 1.5 eV for cases (i) and (ii) respectively. The remarkably distinct potential profiles for the protonated Schiff bases with N=5 in our previous work [Phys. Rev. B 81, 115433 (2010)] and for the N=11 polyene in this work studied by the same GW-BSE approach and code is mostly caused by the different length of the conjugated backbone. Although de Weerd and Ghosh propose their hypothesis, they provide no solid proof and no theoretical calculation to support their hypothesis. Our calculations show that their hypothesis may be incorrect. In fact, in their articles [Chem. Phys. Lett. 354, 38 (2002) and J. Phys. Chem. B 119, 14905 (2015)], they do not exclude the possibility of another dark state near S_2 . They just regard their hypothesis as an alternative explanation to the experimental observations. Please note that our discovery of the S_y state with strong energy shift along the bond length alternation coordinate that crossovers in energy with the S_2 state (Fig. 3) does exhibit a double-well like feature that may explain the spectral shifts observed in these recent experiments.

To give further support to our model, we simulate the 2DES of *Rps. acidophila* which contains RG. The results are shown in Fig. 6 and discussed on Page 7, 8 and 11 in the revised manuscript. Theoretical details are presented on Page 14 and 15. The simulated spectrum is in excellent agreement with the experimental spectrum in Science 340, 52 (2013). Specifically, the diagonal peaks, S_2/S_y cross-peak, and pronounced S_2/S_1 excited-state absorption peak that splits into two by the lower-diagonal S_2/S_y cross-peak are all correctly reproduced in our model simulations. The excellent agreement demonstrates that our theoretical calculations are fully consistent with the 2D experiments.

5. **Comments:** Line 103: *the magnitude of the solution-induced red-shift depends on the solvent thus it can hardly be concluded that it is 0.1 eV, making the*

agreement between calculations and experiment excellent. It will be less than 0.1 eV for hexane and more than 0.1 eV for highly-polarizable solvents such as CS₂.

Reply: We agree with the reviewer that it is not rigorous to simply state that the magnitude of the solution-induced red-shift is 0.1 eV. In the revised manuscript, we remove the sentence “If taking into account the solution-induced ~0.1 eV redshift of the optical absorption spectrum, the agreement between our calculations and experiments is excellent.” and replace it with “**In experiments, the S₂ energy is 2.48 eV for RG in methanol and 2.36 eV for spirilloxanthin in *n*-hexane³. In solution and the protein environment, absorption spectra of Cars redshift owing to the polarizability of the medium^{3, 24, 35, 36}. If taking this into account, our calculations agree well with experiments.**”. Now, the effect of medium polarizability on the redshift is introduced in detail. The S₂ energy for RG from our calculation (2.65 eV) deviates from that measured in methanol (2.48 eV) by 0.17 eV, while the S₂ energy for spirilloxanthin from our calculation (2.42 eV) deviates from that measured in *n*-hexane (2.36 eV) by 0.06 eV. The relatively larger deviation in the former than the latter may be attributed to the higher polarizability of methanol than *n*-hexane.

- Comments:** *Line 109: it is dangerous to compare excited state energies of RG and β -carotene. Although they have formally the same number of conjugated C=C bonds, effective conjugation length of β -carotene is shorter than of RG due to extension of conjugation to terminal rings in β -carotene (see e.g. Fuciman et al. JPCB 2015)*

Reply: We used β -carotene to test the accuracy of our method on the 2A_g⁻ and 1B_u⁻ states energies of RG in the old manuscript since we just considered the total number of conjugated C=C bonds which is 11 for both these two species. According to the suggestion of the reviewer, we calculate the excitation energies of β -carotene and compare them with the β -carotene results from DFT/MRCI in Phys. Rev. Lett. 103, 108302 (2009) (see Supplementary Section 1). The energy differences between states 2A_g⁻, 1B_u⁻ and S₂ from our method compare well with those from DFT/MRCI as can be seen from Supplementary Table S1. This comparison is used just to test the accuracy of our method. It does not affect the conclusion of our work which focuses on the energy difference between the S₂ and S_y states. The original sentence “The values for RG agree with previous work by the multi-reference configuration interaction method (1.99 eV for S₁ and 2.65 eV for 1B_u⁻) for β -carotene with N = 11.” in the old manuscript is removed. Instead, a sentence “**In the Supplementary Section 1, a detailed discussion on the accuracy of our strategy to predict the S₁ and 1B_u⁻ energies is presented.**” is added in the Method section on Page 13.

7. **Comments:** *In line 113 the authors argue that the new state is a singly-excited state, yet it is a combination of two transitions. I believe these two claims contradict each other.*

Reply: These two claims do not contradict each other. We are sorry that we did not explain clearly in the old manuscript that S_y is a singly excited state. The doubly excited states for carotenoids usually refer to the states like $2A_g^-$, $1B_u^-$, etc. which are formed by two coupled triplet excitations. To reach these states from the ground state, two electrons are excited with their spin flipping to create two triplets. For example, $2A_g^-$ is constituted by two T_1 , $1B_u^-$ by T_1+T_2 , where T_1 and T_2 are the lowest and the second lowest triplet excitons.

Both S_2 and S_y states are singly excited state and bear no double excitation character. It is easily to understand that S_2 is a singly excited state since it is formed predominantly by the HOMO→LUMO transition. To explain clearly that S_y is a singly excited state too, Supplementary Section 3 of the revised manuscript discusses in detail the composition and wave function of S_y and other excited states. S_y is composed dominantly by the HOMO→LUMO+1 and HOMO-1→LUMO transitions. These two components do not mean that one electron is promoted from HOMO to LUMO+1 and another from HOMO-1 to LUMO, and do not mean that S_y is a double excitation. It only means that the wave function of the S_y exciton can be represented by the linear combination of $\psi_{H-1}(\mathbf{r}_h)\psi_L(\mathbf{r}_e)$ and $\psi_H(\mathbf{r}_h)\psi_{L+1}(\mathbf{r}_e)$ in the language of BSE, where \mathbf{r}_h and \mathbf{r}_e are coordinates of the hole and excited electron respectively (see Supplementary Section 3 for details), and spatial distribution of the hole (the excited electron) can be described by the combination of HOMO-1 and HOMO (LUMO and LUMO+1). Above S_y there is another state which is also composed by HOMO→LUMO+1 and HOMO-1→LUMO but with different phase from S_y . This state can be attributed to the singly excited $1A_g^+$ state which has been well defined in the community (see Supplementary Section 3 for details).

Above all, the combination of two transitions does not contradict singly-excited character of the S_y state.

8. **Comments:** *In their arguments, the authors obviously mix two different states discussed in literature. While they try solve the "Bu- problem", references 8 and 25 deals with another state, tentatively denoted as the S^* state. I admit that in the first studies of the S^* state it was speculated that it might be the Bu- state, but later it was clearly shown that S^* state (Refs 8 and 25) has nothing common with the Bu- state.*

Reply: We would like to thank the reviewer to remind us this error in the old manuscript. In the revised manuscript, (i) Ref. 8 is removed from the original sentence “This dark state plays an critical role in mediating the Car-to-Chl energy transfer process and the depopulation of the S₂ state^[3, 5-12].” which is on Page 2; (ii) the words “agreeing with the experiments^[8, 25]” is removed from the original sentence “The overall S_y-mediated energy transfer rate is one order of magnitude smaller than that of the S₂-mediated one, agreeing with the experiments^[8, 25].” on Page 10; (iii) sentences “**Dark states play key roles in the excited-state dynamics of Cars. The feature of some dark states has been defined. For example, the S* state, which is close in energy to the lowest singlet excited state S₁, is the precursor to the formation of triplet states⁵⁻⁷.**” are added in the Introduction Section on Page 2. These revisions do not influence our conclusions.

9. **Comments:** *Line 208: could the authors provide a reference to "experimental RG B800 spectrum"? I do not recall that such spectrum has ever been measured experimentally.*

Reply: In the old manuscript, we stated that “Fig. 6a shows the calculated absorption spectrum of the RG-B800 complex which is comparable to the experimental spectrum.”. We said this since our theoretical spectrum reproduces the B_x, S₂, Q_x and Q_y peaks in the experimental spectra from the references (i) Science 340, 52 (2013) for the LH2 complexes of *Rps. acidophila* (Figure 1 therein) and (ii) Q. Rev. Biophys. 39, 227 (2006) for the isolated *Rps. acidophila* membrane containing LH2 complexes (Figure 3 and Figure 5 therein). We are sorry for the misleading caused by the simplification in our description. In the revised manuscript we modify this sentence to “**Fig. 7a shows the calculated absorption spectrum of the RG-B800 pairs which is comparable to the experimental spectrum of LH2 complex in *Rps. acidophila* strain 10050^{9, 40}.**” in Page 9. References 9 and 40 are the two articles mentioned above. Please note that the original Fig. 6 is now Fig. 7 in the revised manuscript. A new figure is added as Fig. 6.

10. **Comments:** *The excitation and emission energy of the Bu- state is quite complicated issue that is far from being settled. The "Bu- emission" was reported only by Koyama group (Ref 19, not 18 as incorrectly stated in line 65) and the energies extracted from data in Fig. 19 are based solely on fitting and it is not clear whether the data contains any Bu- emission at all. Thus, I would be careful to claim that Bu- emission energy is less than 2 eV.*

Reply: Emission energy of the 1B_u⁻ is widely discussed in the review article by Polívka and Sundström in Chem. Rev. 104, 2021 (2004). Description of the 1B_u⁻

emission energy in our manuscript is based on this review. According to the reviewer's suggestion, the following changes are made to the manuscript:

(i) The original sentence "This challenges the ascription of the dark state to the $1B_u^-$ state since the emission energy of the $1B_u^-$ state is smaller than 2 eV for these three Cars^[3, 18], for example the $1B_u^-$ energy of spirilloxanthin from the experimental spectrum is just 1.65 eV^[3, 18]." is changed to "In addition, emission energy of the $1B_u^-$ state might be smaller than 2 eV for Cars with N=10~13 as predicted by Koyama *et al*^{3, 20}, according to which energy transfer to the Chls Q_x state, whose absorption energy is ~2.1 eV, cannot realize." on Page 3. Ref. 20 is the original Ref. 19. The original Ref. 18 also discusses the $1B_u^-$ emission energy in Figure 5 therein. The original Ref. 19 discusses the $1B_u^-$ emission energy specially. In the revised manuscript, the original Ref. 19 is cited here instead.

(ii) The sentence "This is different from the $1B_u^-$ state whose $0\leftarrow 0$ emission energy is already below the energy of the S_2 state by 0.4 eV for N = 11 and their gap increases further to 0.6 eV for N = 13 according to experimental spectra^[8]. The too low potential minimum of the $1B_u^-$ state for Cars with N \geq 11 (< 2.0 eV) (Fig. 5b) excludes the possibility of it to be the origin of the experimental dark state." is removed in the revised manuscript.

All these modifications do not affect our conclusions.

Reviewer #3:

Comments: *This paper addresses a long-standing topical issue regarding the electronic states of carotenoids and their roles in light harvesting. The main advance of the paper is the improved electronic structure calculations compared to previous studies. The conclusion of the paper suggests that this work is the last word on the subject, which I think is misleading and overstates the advance and its generality. Nevertheless, the work is clear and is a significant result that is well suited for publication in this journal.*

Reply: We appreciate the reviewer's support for the publication of our work. According to the reviewer's comments, we rewrite the conclusion section of the manuscript to avoid misleading. The conclusion section is now changed to "In conclusion, we find a new state S_y of the A_g^+ symmetry in Cars which may account for the origin and nature of the controversial dark intermediate state

between the S_1 and S_2 states. Its energy transfer capability and its conjugation length dependence of excited-state dynamics studied by many-body Green's function theory and simulated 2D electronic spectrum indicate that the S_y state could play an important role in the energy migration in photosynthesis. This new state could improve our conventional understandings on the electronic structures of Cars and is helpful to explore the early events in photosynthesis." Certainly, the conclusions of our work do not change. Furthermore, the new simulated 2D electronic spectrum and other additional calculations further support our conclusions.

1. **Comments:** *The precise nature and assignment of S_y is not completely clear to me. Is this likely to be the "x" state that was observed by Ostroumov et al in their 2D spectroscopic studies of LH2?*

Reply: We revise the manuscript greatly and describe the S_y state in more details. Especially, in the Supplementary Section 3 we analyze the wave function, composition, symmetry and oscillator strength of the S_y state and compare these features with those of other excited states. The main features of the S_y state is listed below:

- (i) It is a singly excited state. We have to stress this point since it may be falsely regarded as a doubly excited state according to its composition. It is formed by promoting one electron from the S_0 state, not two electrons like the doubly excited $2A_g^-$ and $1B_u^-$ states.
- (ii) Its symmetry is A_g^+ . Its composition resembles that of the well-known $1A_g^+$ state, both composed mainly by the HOMO \rightarrow LUMO+1 and HOMO-1 \rightarrow LUMO transitions. The distinction between the S_y state and the $1A_g^+$ state lies in the different phase in their wave functions. Details can be found in Supplementary Section 3.
- (iii) It remains optically forbidden for both the all-*trans* and *cis* configurations, while the $1A_g^+$ state is optically forbidden in the all-*trans* configuration but turns to be allowed in the *cis* configuration.
- (iv) Its absorption energy is ~ 0.5 eV higher than the S_2 state, but its emission energy is a little bit (~ 0.1 eV) lower than the S_2 state. The potential surfaces of the S_2 and S_y states form a double-well structure (see Fig. 3, Supplementary Figure S7 and Figure S8), making nonadiabatic transition from S_2 to S_y realizable.

According to the conjugation length dependence of the emission energy gap between the S_2 and S_y states, we think that the S_y state can be assigned to the "X" state that was observed by Ostroumov *et al.* in their 2D spectroscopic studies of LH2. To further verify this conclusion, we simulate the 2DES of *Rps. acidophila* which contains RG. The results are shown in Fig. 6 and discussed on Page 7, 8

and 11 in the revised manuscript. Theoretical details are presented on Page 14 and 15. The simulated spectrum is in excellent agreement with the experimental spectrum in the work by Ostroumov *et al.* Specifically, the diagonal peaks, S_2/S_y cross-peak, and pronounced S_2/S_1 excited-state absorption peak that splits into two by the lower-diagonal S_2/S_y cross-peak are all correctly reproduced in our model simulations. The excellent agreement demonstrates that our theoretical calculations are fully consistent with the 2D experiments and the S_y state is responsible for the X state.

REVIEWERS' COMMENTS:

Reviewer #1 (Remarks to the Author):

The authors made significant efforts in the new version of the manuscript. They improve the quality of their work with additional data and additional clarifications. My main technical concern (which was also shared by Reviewer #2) turned out to be just a missing clarification. I believe that in the present form the conclusion of the manuscript are now sufficiently supported by the reported data. Moreover, in the new conclusions the authors are correctly less categorical and conclusive (issue that was also suggested by Reviewer #3). Therefore, in my opinion, the manuscript deserves publication in this journal.

Reviewer #2 (Remarks to the Author):

The authors have clearly invested a lot of effort to clarify the points raised by the reviewers and I agree that the revised version is significantly improved. Introduction of a new state into the carotenoid excited-state manifold is certainly a shift in our understanding of carotenoid excited states and yes, I admit that the revised version provides likely enough evidence to make such a claim. Only future experiments may show whether the calculations reported in the manuscript are accurate enough. Since the conclusions of this manuscript (regardless whether they are correct or not) will surely initiate further experimental and theoretical studies, this hypothesis of a new state with Ag+ symmetry should be published. Prior to publication though, there are still some issues that should be taken care of:

1. As in my report to the original manuscript, I must note that the authors have some gaps in their understanding carotenoid photophysics. In the new text in Introduction, lines 43-46 they write that "the S* state that is close in energy to the lowest singlet excited state S1, is the precursor to the formation of triplet states". The issue of the S* state is far from being settled, recent evidence actually points rather to a hot ground state as the origin of the S* signal, and formation of triplet from S* has never been observed for carotenoids in solution. Thus, it is actually not known at all what is the energy of the S* state and whether it is close to the S1 or not. Since this manuscript do not deal with S* at all, I would recommend to remove this part from Intro completely as it only confuses the potential readers.
2. In the new text in lines 334-342 there are two claims that are clearly overstatements and should be modified or removed.

First, the authors claim that the new Sy state resolves the problem of the controversial dark intermediate state. Yet, in Fig. 3 they still have the Bu- state (that was the main subject of this controversy) within the S2-S1 energy gap. So, what about the Bu- state in their model? Is it involved in relaxation or not? Can these new calculations help to resolve the Bu- problem?

Second, and probably more importantly, they claim the Sy state "could play an important role in energy migration in photosynthesis". What do they mean by the word "important"? In LH2 from *Rps. acidophila*, the S2 (Bu+) state transfer energy to BChl_a with about 50% efficiency and sub-100 fs rate. If the Sy-Qx transfer has a rate an order of magnitude slower (lines 267-268), it means that the efficiency of the transfer from the Sy state will be for sure less than 10% - where is the importance then? The contribution of the Sy transfer to the overall energy transfer is then rather negligible and certainly cannot affect the fitness of the organism. I agree that this channel might be there but to claim that it is important is clearly an overstatement? Important for what?

Reviewer #3 (Remarks to the Author):

The revised paper is much stronger. In particular the amazing agreement between the simulated and previously published experimental 2D spectrum is compelling. I think this work will be very helpful for the field and strongly recommend publication.

Reviewer #1 (Remarks to the Author):

Comments: *The authors made significant efforts in the new version of the manuscript. They improve the quality of their work with additional data and additional clarifications. My main technical concern (which was also shared by Reviewer #2) turned out to be just a missing clarification. I believe that in the present form the conclusion of the manuscript are now sufficiently supported by the reported data. Moreover, in the new conclusions the authors are correctly less categorical and conclusive (issue that was also suggested by Reviewer #3).*

Therefore, in my opinion, the manuscript deserves publication in this journal.

Reply: We thank the reviewer very much for approving our manuscript to be published in Nature Communications.

Reviewer #2 (Remarks to the Author):

Comments: *The authors have clearly invested a lot of effort to clarify the points raised by the reviewers and I agree that the revised version is significantly improved. Introduction of a new state into the carotenoid excited-state manifold is certainly a shift in our understanding of carotenoid excited states and yes, I admit that the revised version provides likely enough evidence to make such a claim. Only future experiments may show whether the calculations reported in the manuscript are accurate enough. Since the conclusions of this manuscript (regardless whether they are correct or not) will surely initiate further experimental and theoretical studies, this hypothesis of a new state with A_g^+ symmetry should be published. Prior to publication though, there are still some issues that should be taken care of:*

Reply: We thank the reviewer very much for approving our manuscript to be published in Nature Communications.

Comment: *1. As in my report to the original manuscript, I must note*

that the authors have some gaps in their understanding carotenoid photophysics. In the new text in Introduction, lines 43-46 they write that "the S state that is close in energy to the lowest singlet excited state S1, is the precursor to the formation of triplet states". The issue of the S* state is far from being settled, recent evidence actually points rather to a hot ground state as the origin of the S* signal, and formation of triplet from S* has never been observed for carotenoids in solution. Thus, it is actually not known at all what is the energy of the S* state and whether it is close to the S1 or not. Since this manuscript do not deal with S* at all, I would recommend to remove this part from Intro completely as it only confuses the potential readers.*

Reply: We would like to thank the reviewer to remind us of this point related to the S* state. According to the reviewer's suggestion, we remove this part from the manuscript to avoid confusion.

Comment: 2. *In the new text in lines 334-342 there are two claims that are clearly overstatements and should be modified or removed. First, the authors claim that the new Sy state resolves the problem of the controversial dark intermediate state. Yet, in Fig. 3 they still have the Bu- state (that was the main subject of this controversy) within the S2-S1 energy gap. So, what about the Bu- state in their model? Is it involved in relaxation or not? Can these new calculations help to resolve the Bu- problem? Second, and probably more importantly, they claim the Sy state "could play an important role in energy migration in photosynthesis". What do they mean by the word "important"? In LH2 from *Rps. acidophila*, the S2 (Bu+) state transfer energy to BChl_a with about 50% efficiency and sub-100 fs rate. If the Sy-Qx transfer has a rate an order of magnitude slower (lines 267-268), it means that the efficiency of the transfer from the Sy state will be for sure less than 10% - where is the importance then? The contribution of the Sy transfer to the overall energy transfer is then rather negligible and certainly cannot affect the fitness of the organism. I agree that this channel might be there but to claim that it is important is clearly an overstatement? Important for what?*

Reply: First, we think that it is the new A_g⁺ state S_y, not the 1B_u⁻ state,

that is responsible for the controversial dark intermediate state. This is the main conclusion of our paper. However, we do not exclude the existence of the $1B_u^-$ state within the S_2 - S_1 energy gap, according to discussions in previous literatures and our present calculations. We include the $1B_u^-$ state in Fig. 3 to give a more complete picture of the carotenoid photophysics. Since the $1B_u^-$ state is beyond the capability of our theoretical methods, our calculations cannot solve the $1B_u^-$ problem and we do not know whether $1B_u^-$ is involved in the relaxation or not. However, this does not influence the conclusions of our paper.

Second, we think that the importance of the S_y state in the energy migration in photosynthesis is due not only to the direct energy transfer from it to chlorophyll, but also to the role it plays as the intermediate state in modulating the decay from the S_2 to S_1 states as observed in experiments [*Science* 298, 2395-2398 (2002) and *Science* 340, 52-56 (2013)]. To avoid confusion to the readers, the lines 334-342 are modified according to the reviewer's comments and the editor's suggestion as following

'In conclusion, our study provides evidence for a new state S_y of the A_g^+ symmetry in Cars, thus contributing to a better understanding of carotenoid excited states. Future experiments would be required to test the accuracy of our calculations.'

Reviewer #3 (Remarks to the Author):

Comment: *The revised paper is much stronger. In particular the amazing agreement between the simulated and previously published experimental 2D spectrum is compelling. I think this work will be very helpful for the field and strongly recommend publication.*

Reply: We thank the reviewer very much for approving our manuscript to be published in Nature Communications.